Deep reinforcement learning models in auction item price prediction: an optimisation study of a cross-interval quotation strategy

http://orcid.org/0000-0002-8299-8278 Ke Da 1
Fan Xianhua 2 Fanxh@cug.edu.cn
1 School of Management, Huazhong University of Science and Technology , Wuhan, Hubei , China
2 School of Economics and Management, China University of Geosciences , Wuhan, Hubei , China
Aleem Muhammad
Electronic publication date: 2024 Jul 11
Publication date: 2024
Volume: 10
Electronic Location ID: e2159
Received 2024 Mar 29; Accepted 2024 Jun 6
Copyright: © 2024 Ke and Fan
Copyright year: 2024
Copyright holder: Ke and Fan
License: This is an open access article distributed under the terms of the Creative Commons Attribution License, which permits unrestricted use, distribution, reproduction and adaptation in any medium and for any purpose provided that it is properly attributed. For attribution, the original author(s), title, publication source (PeerJ Computer Science) and either DOI or URL of the article must be cited.
License URL: https://creativecommons.org/licenses/by/4.0/

Keywords: Auction price, GRU, Reinforcement learning, Price interval estimation

Funding: The authors received no funding for this work.

==============================
In the contemporary digitalization landscape and technological advancement, the auction industry undergoes a metamorphosis, assuming a pivotal role as a transactional paradigm. Functioning as a mechanism for pricing commodities or services, the procedural intricacies and efficiency of auctions directly influence market dynamics and participant engagement. Harnessing the advancing capabilities of artificial intelligence (AI) technology, the auction sector proactively integrates AI methodologies to augment efficacy and enrich user interactions. This study delves into the intricacies of the price prediction challenge within the auction domain, introducing a sophisticated RL-GRU framework for price interval analysis. The framework commences by adeptly conducting quantitative feature extraction of commodities through GRU, subsequently orchestrating dynamic interactions within the model’s environment via reinforcement learning techniques. Ultimately, it accomplishes the task of interval division and recognition of auction commodity prices through a discerning classification module. Demonstrating precision exceeding 90% across publicly available and internally curated datasets within five intervals and exhibiting superior performance within eight intervals, this framework contributes valuable technical insights for future endeavours in auction price interval prediction challenges.

Introduction

An auction, an intricate mechanism for ascertaining the value of commodities or services, unfolds through the orchestration of bids and offers. In this dynamic setting, sellers provide their goods or services, employing bidding or pricing strategies. Concurrently, buyers articulate their demand through bidding or quoting, culminating in elevating the highest bidder or the most favourable offer, thereby securing the privilege to engage in the trade. This transactional paradigm finds versatile application across various domains, such as artwork, real estate, and agricultural products (Rashed, Jawed & Schmidt-Thieme, 2020). The evolution of auctions traces a developmental trajectory from conventional live auctions to the contemporary realm of online auctions. Propelled by technological advancements, the ubiquity of the Internet, and the flourishing digital technology landscape, individuals can now engage in global auction activities through online platforms. This transformative shift enhances the breadth of participation and the transparency inherent in the auction market (Hu, Dowd & Bouchard, 2022).

In artificial intelligence (AI), the rapid advancement of technology has positively impacted the development of auction systems. Firstly, integrating intelligent recommendation systems and advanced data analysis technologies is pivotal in endowing auction platforms with an enhanced understanding of user requirements. This, in turn, facilitates bespoke auction services, fostering heightened user engagement. Secondly, deploying machine learning algorithms proves instrumental in price prediction and risk management, amplifying the precision of auction pricing and mitigating transactional risks (Luong, Jiao & Wang, 2020). In price prediction, support vector machines (SVM) unravel intricate nonlinear relationships within historical auction data, enabling the anticipation of commodity prices. Concurrently, deep learning regression models, exemplified by neural networks, adeptly discern the intricate patterns underlying price fluctuations through a hierarchical learning structure, elevating sensitivity to market dynamics. Delving into personalised recommender systems, algorithms such as multilayer perceptrons (MLPs) or sequential models like long short-term memory networks (LSTMs) meticulously navigate temporal relationships within user behaviors. This nuanced approach enhances comprehension of user preferences, facilitating the recommendation of commodities aligned with individual user interests (Rhee, Ahn & Oh, 2021).

Simultaneously, the conduct of auctions can be construed as a financial market trading endeavour, wherein the outcome is intricately linked to the external milieu. In contrast to models grounded in machine learning, reinforcement learning emerges as a potent approach, leveraging information and feedback gleaned directly from the financial market environment. This process culminates in decision-making orchestrated through trading agents, who craft astute trading strategies. Deep reinforcement learning, harnessing the prowess of deep learning techniques, adeptly extracting robust features from raw data, and enabling the discernment of external market dynamics. This method, characterized by its integration of reinforcement learning, proves well-suited to the cacophony of financial and external market landscapes, constructing strategies that maximise rewards (Qiu, Wang & Wang, 2021). Noteworthy deep reinforcement learning frameworks, such as Deep Q Networks (DQN), Asynchronous Dominant Action Evaluation (A3C), and Deep Deterministic Policy Gradient (DDPG), find frequent applications. The performance of trading agents constructed within these frameworks surpasses that of a singular reinforcement learning framework. Consequently, this study delves into the intricacies of the inter-area offer prediction challenge within the auction process. It aims to furnish intelligent methodological support for the auction market, with the ensuing contributions outlined in the subsequent discourse. 1. A crawler method is used to gather online auction data for digital products, followed by detailed feature refinement based on relevant research.

2. Feature extraction is enhanced using the Gated Recurrent Unit (GRU) approach, and the RL-GRU model is developed by integrating reinforcement learning with a Deep Q Network (DQN) to predict auction price ranges.

3. The model’s effectiveness in predicting auction prices is evaluated across various data sizes, using both public and private datasets, showing superior performance over traditional classification methods.

The rest of the article is organised as follows: “Related Works” introduces the related works on auction analysis and reinforcement learning. The proposed framework, RL-GRU is established in “Methodology”. “Experiment Result and Analysis” describes the experiment setup and results, and a conclusion is drawn at the end.

Related works

Studies on the impact and forecasting of auction prices

Lucking-Reiley et al. (2004) delved into one-cent collectible coin auctions on eBay, amassing a substantial dataset. Their findings underscored the impact of the seller’s credit rating, starting price, reserve price, and auction duration on the ultimate auction price. Depken & Gregorius (2010), in an exploration of eBay auctions, scrutinised the influence of auctioneer reputation, auction timing, the aesthetics of the auction page, and other factors on transaction prices. Hou (2007) conducted a comparative analysis of auction data about 17–19-inch LCDs in eBay China and the U.S., revealing similarities and disparities in factors affecting online auction prices across the two countries. While starting price and seller’s reputation exhibited similar effects, variations in the strength of these factors were noted in culturally diverse nations.

Similarly, Wood (2001) scrutinised 7,362 transactions of collectible coins on eBay, identifying the effects of weekend, picture, prestige, and auction time as pivotal factors shaping auction outcomes. Kauffman & Wood (2006) delved into bidders’ willingness to pay, discovering that auctions concluded on weekends, items with images, and items from highly creditworthy sellers elicited higher prices. Budish & Takeyama (2001) examined sellers’ expected returns in first-priced closed English auctions, proposing a one-bite pricing strategy to attract risk-averse bidders and enhance sellers’ profits. Melnik & Alm (2002) investigated the impact of sellers’ credit on buyers’ willingness to pay, utilising data from eBay auctions of mint condition $5 gold coins. The empirical results affirmed that the quality of sellers’ credit significantly influenced auction prices.

Moreover, Ghani & Simmons (2004) developed a predictive mechanism for online auction final prices, incorporating textual information provided by sellers as attributes and employing a binary classification method. Wang, Jank & Shmueli (2008) constructed a dynamic prediction model for ongoing auctions, continuously updating price predictions with new information. Despite utilising generalised data analysis methods, the results obtained were deemed unsatisfactory. This highlights the inherent challenge of achieving satisfactory outcomes through reliance on a singular data model for the prediction and analysis of auction processes.

From the above mentioned research, it can be seen that in current auction price analysis research, the main direction of auction-related prediction research is to use the information on the product itself and historical transaction information to analyse the rationality and risk of transactions. Simultaneously investigating the characteristics of different attributes is also a focus of such research, but predicting the final auction price by analysing each attribute and historical data is limited by the amount of data and the performance of the analysis model, making it difficult to achieve price prediction of auction products. This is an important issue that urgently needs to be addressed to develop intelligent auctions.

Deep reinforcement learning methods in financial class prediction

Auctions, inherently intertwined with finance, find manifestations in asset auctions, enterprise financing auctions, and risk management. To enhance current research on auction price intervals, insights from deep learning and reinforcement deep learning in the financial domain can be judiciously incorporated. Carapuço, Neves & Horta (2018) ingeniously merged a three-layer neural network with the Q-learning method, crafting a high-frequency trading system for the foreign exchange market. The stability of the trading agent training system in dataset training was affirmed, enabling sound out-of-sample trading decisions. Li, Ni & Chang (2020) introduced a deep reinforcement learning strategy for stock trading, comparing classical deep reinforcement learning (DRL) models (DQN, DDQN, and Duelling DDQN) with DQN exhibiting superior out-of-sample trading decision capabilities. A hybrid solution was proposed, integrating an improved DRL method and the Adaboost algorithm.

Likewise, Sornmayura (2019) evaluated the performance of a trading agent’s strategy based on the DQN method against buy-and-hold and Commodity Trading Advisor (CTA) strategies, demonstrating its superiority in the foreign exchange market for the euro and the U.S. dollar. Singh & Srivastava (2016) employed deep neural networks for feature extraction, utilising an LSTM-based reinforcement algorithm for trading decisions in the futures market. The application of a multi-objective structure with varying weights for profit and risk balancing revealed an intuitive and profitable strategy. Furthermore, Zarkias, Passalis & Tsantekidis (2019) introduced a price-tracking method, employing a trading agent based on deep learning to track prices within a specified boundary. This approach enhanced the trading agent’s resilience to noise and proficiency in identifying price trends. Jeong & Kim (2019) integrated a deep neural network into a reinforcement learning algorithm for stock number prediction, analyzing stock prices based on diverse action curves derived from the price. Transfer learning was incorporated to mitigate overfitting concerns. Moreover, Liang et al. (2018) implemented successive reinforcement learning algorithms (deep deterministic policy gradient (DDPG), proximal policy optimization (PPO), policy gradient (PG)) for portfolio management in the Chinese stock market. An adversarial training method significantly improved training efficiency and backtested average daily returns and Sharpe ratios.

Incorporating deep learning into the financial sector has profoundly transformed how data is processed, particularly with its robust handling of multimodal and dynamic data, a common feature in economic and financial contexts. Deep learning excels in integrating diverse data types, which is crucial for applications where traditional methods struggle, such as in rapidly changing auction markets. Here, the dynamic nature of auction data, characterised by rapid fluctuations and complex price influences, makes deep learning particularly beneficial.

However, significant challenges persist in the application of these technologies. Deep learning models depend heavily on large volumes of high-quality data, which are not always available due to privacy concerns and the rarity of specific financial data sets. Additionally, the complexity of these models often leads to a lack of transparency in decision-making, a critical issue in financial contexts where explainability is essential for trust and regulatory compliance. These challenges highlight the need for ongoing research to refine these models for practical, real-world application in the financial domain.

Methodology

GRU

A recurrent neural network (RNN) is a category of neural networks designed for processing sequential data. In contrast to traditional neural networks, RNNs feature recurrent connections that enable them to model and analyse sequential information. The Gated Recurrent Unit (GRU) represents a specific variant of RNN tailored for the processing and learning of sequential data.

The primary objective behind the design of GRU is to address the issue of vanishing gradients prevalent in traditional RNNs, simultaneously reducing the number of parameters and enhancing the model’s training efficiency. GRU incorporates two essential gating mechanisms: the Update Gate and the Reset Gate. These mechanisms enable the selective updating of information during the processing of sequence data, thereby improving gradient flow and facilitating the capture of long-term dependencies (Cahuantzi, Chen & Güttel, 2023). The typical structure of a GRU cell is depicted in Fig. 1:

Figure 1 The structure of the GRU cell.

The dashed lines are labeled with their special update gate and overlap gate structures, both of which are calculated by Eqs. (1) and (2):

(1) zt=σ(Wz⋅[ht−1,xt])

(2) rt=σ(Wr⋅[ht−1,xt])

where zt is the output of the update gate, and rt is the reset gate output. σ is the Sigmoid activation function, W is the weight matrix, and ht−1 is the hidden state of the previous moment, and xt is the input of the current moment.

Update gates and reset gates are two key components in GRU, and by introducing a gating mechanism, they control the extent to which the model updates the hidden state at the current moment and retains the information of the previous moment when calculating candidate hidden states, respectively. The update gate enables the model to selectively retain or update historical information, which helps to deal with long-term dependencies. In contrast, the reset gate enables the model to dynamically choose to retain or ignore information from the previous moment, adapting to changes and different patterns in the sequence, which together improves the performance of GRU in sequence data processing. After going through the settings of the reset gate and update gate, the enhancement of the time series information can be accomplished by calculating the hidden state and updating it, the process of which is shown in Eqs. (3) & (4):

(3) h~ t=tanh(Wh⋅[rt⊙ht−1,xt])

(4) ht=(1−zt)⊙ht−1+zt⊙h~ t

where ⊙ denotes the element-by-element multiplication, and h~ t is the candidate’s hidden state, the ht the final hidden state and tanh is the hyperbolic tangent activation function, and Wh is the weight matrix. The combination of Eqs. (3) and (4) reveal that the final state ht is the update gate controlled by the previous moment’s hidden state ht−1 and the candidate hidden state which are linear combinations of the previous moment’s hidden state and the candidate hidden state under the control of the update gate. With the above state update, the feature enhancement and extraction of the current data can be realised.

In GRU, the recurrent unit processes input information, computes the state output of the hidden layer at the current time step, and subsequently, through the neural network output layer calculation, derives the final result of the entire model. The formulation for the output layer can be expressed as follows:

(5) nety=wyht

(6) y=σ( net y)

where wy is the weights, and y is the output of the neural network, nety is the input of the excitation function. The model completes the updating of its own weights by gradient descent during the training process.

Reinforcement learning and DQN

Reinforcement learning (RL) constitutes a machine learning paradigm that empowers an intelligent agent to acquire decision-making prowess in a given task. This is achieved through the agent’s interaction with the environment, aiming to maximise the cumulative reward signal. In reinforcement learning, the intelligent agent undergoes a learning process through trial and error, without explicit data labeling. Instead, it gains experiential knowledge by actively engaging with the environment. The Q-learning method stands out as a value iteration algorithm among the widely employed reinforcement learning methods. Distinguishing itself from the conventional value iteration approach, Q-learning emphasises estimating state-action value pairs rather than solely focusing on state value estimation. This method is particularly concerned with determining the value associated with different actions in a given state (Li, Chen & Zhao, 2022).

Building on this foundation, the Q-learning algorithm defines an action (state-action) value function, denoted as Q* (s, a), which signifies the value associated with different actions in a specific state under a given strategy x. Specifically, Q* (s, a) is the expected reward for taking action ‘a’ in state ‘s’ under strategy x’s influence. It’s important to note that the rewards in the action value function, denoted as r, differ from those in the state value function. In the context of the action value function, the reward (r) pertains to the outcome after an action is executed by an intelligent agent. Conversely, in the state value function, the reward corresponds to the expected value of rewards across multiple actions.

Consequently, Qπ (s, a) serves as a metric for evaluating the desirability of a particular action in a given state. This function is more intuitive and comprehensible (Fan, Wang & Xie, 2020). The action value function can be expressed in terms of the state value function, as depicted in Eq. (7).

(7) Qx(s,a)=Eγ−S[rs,a+Ws′]=∑pa,s→s′(rs,a+γVs).

Qs,a is equal to the sum of the desired immediate reward and the s′ the sum of the long-term discounted rewards of the state. The optimal Q∗(s,a) can be written as:

(8) Q∗(s,a)=rs,a+γmaxQ(s′,a′).

At the same time, the optimal state value function V∗(s) and the optimal action value function Q∗(s,a) are related as follows:

(9) V∗(s)=maxQ∗(s,a)

The optimal policy at this point can be calculated by Eq. (10):

(10) π∗(s)=argmaxQ∗(s,a).

However, in the actual solution process, the full updated Q-value that needs to be known at the time of iteration cannot be obtained directly, so in practical applications, Eq. (8) can be used by Eq. (11) shown in the

(11) Q∗(s,a)=Q(s,a)+α[r+γmax,⁡Q(s′,a′)−Q(s,a)]

where α is the learning rate; from the above equation, we can get the optimal solution of Q∗(s,a) Deep Reinforcement Learning (DRL) is an algorithmic approach that amalgamates deep learning with reinforcement learning, enabling end-to-end learning from perception to action. This methodology mirrors the human learning process, where environmental information is inputted, deep neural network training is undertaken, and the output directly corresponds to actions.

In the realm of value function-based reinforcement learning, the need to generalise the estimation of state-action sequences prompts the utilisation of a “function approximator.” This approximator, often a linear function, is employed to estimate the value function associated with state-action pairs. This approach involves using a function to approximate the distribution of values. The formula below illustrates this, where ‘f’ represents the function approximator, and ‘ω’ uniformly signifies the parameters utilised in the function (Carta, Ferreira & Podda, 2021).

(12) Q(s,a)≈f(s,a,ω).

This process in deep reinforcement learning can be accomplished by the DQN network, and its overall updating process is shown in Fig. 2.

Figure 2 The framework for the DQN.

As depicted in Fig. 2, the deep network is instrumental in selecting the optimal reward value based on the current state to determine whether a corresponding action should be taken. This action then interacts with the environment to acquire feedback on the reward value. Subsequently, in the subsequent moment, the intelligent agent recommences the process by perceiving the new state, taking a new action, and initiating a fresh round of training. This iterative cycle persists until the completion of the entire training process.

RL-GRU framework establishment

Following the exposition on GRU and reinforcement learning, the model training and classification of price intervals were implemented based on the data features. The comprehensive framework of this implementation is illustrated in Fig. 3.

Figure 3 The framework for the RL-GRU.

In this framework, the relevant characteristics of the product were first quantified to predict the price range. Specifically, we numerically characterised the product’s key features and constructed an output vector X = {x1, x2,…, xn}. We selected ten basic data items for in-depth analysis of bidding products and completed the construction of the input layer. Furthermore, we applied a GRU to mine data features deeply. The gating mechanism of GRU optimises temporal data processing, effectively capturing long-term dependencies and more accurately depicting the dynamic trend of commodity prices. This stage provides rich feature representations for the model and lays a solid foundation for learning tasks. The key improvement is the introduction of reinforcement learning techniques, especially Q-learning. We used Deep Reinforcement Learning Network (DQN) to train the model. DQN continuously adjusts its decision-making strategy through interactive learning with the environment to maximise cumulative rewards. This method significantly improves the model’s adaptability to dynamic changes in the auction market and enhances its robustness in complex environments. The introduction of reinforcement learning enables the model to learn and optimise its strategies from past interactions. It enhances its predictive performance, which is crucial for predicting the price range of auction products. In the end, the comprehensive application of these technologies successfully predicted the price range of auction products.

Experiment result and analysis

Dataset and training process

Given the substantial impact of price fluctuations in online auctions and the variability in product attributes, this study employs crawling methods to analyse pertinent auction data on the eBay website. The data construction integrates insights from Elshaar & Sadaoui (2019) and Alzahrani & Sadaoui (2018) encompassing attributes such as goods details, pricing, auction timing, postage, and ten indices. This article has collected over 3,000 pieces of eBay transaction data, including the required information. Based on this content, data partitioning has been completed. In order to better study product performance and distinguish auction product intervals, this article mainly analyses digital camera data, making it applicable to the reference dataset. Its adequate sample size exceeds 1,000.

Data is obtained in the data screening process by selecting the brand with the highest transaction volume. Following the construction of the dataset, the model is trained and analysed. Price interval division is conducted into three intervals, five intervals, and eight intervals, with the calculation of interval ranges accomplished using Eq. (13):

(13) Pr=(Max−Min)/n

where pr represents the price range for the final prediction object, max represents the highest price of the category, min is the lowest price, and n is 3,5,8 three price intervals. The ultimate objective of this manuscript culminates in the meticulous demarcation of the final price spectrum, thereby transmuting its essence into a discernible class of identification quandaries. Consequently, in the evaluative crucible of our model, precision, recall, and the F1-score stand as arbiter metrics. Their scrutiny reveals the intricacies of the computational tapestry encapsulated within Eqs. (14)–(16);

(14) P=TPTP+FP

(15) R=TPTP+FN

(16) F1=2×P×RP+R

where TP is the true positive, FP is the false positive and FN represents the false negative. Based on the correlations that have been defined and the comparison metrics that have been determined, we have obtained the model training procedure for this article as shown in Algorithm1.

Algorithm 1 Training process of RL-GRU

Input: Auction characteristic from eBay dataset	
Initialization.	
Define the RL-GRU	
Define the training information including: initial parameters, optimiser and max training epochs.	
Feature extraction.	
Using the GRU to extract the commodity feature	
Model training: Epochs initialization.	
while epoch< set preset epoch do	
Sample data from Input.	
Feed data to RL-GRU.	
Model updates.	
End	
Parameters optimisation	
while epoch< preset epoch do	
Feed validation data to commodity.	
Loss calculation.	
Compute precision, recall and F1-score	
Save the optimal model	
end	
Output: Trained RL-GRU	

The architecture and the specific information for the network is given as Table 1.

Table 1 The architecture for the network and hyperparamets.

Number of layers	3	
Activation function	‘ReLU’	
Learning rate	0.01	
Discount factor	0.95	
Exploration rate	0.1	
Batch size	32	
Epochs	100	
Optimizer	‘Adam’	
Loss function	‘MSE’	

Experiment result

Upon the consummation of the construction and training of our model, meticulous analysis and appraisal of the data ensued. Drawing from the eBay’s repository (https://zenodo.org/records/8126348), the data under scrutiny spans various auction price ranges in the model’s testing phase. Comparative juxtapositions with conventional methodologies such as SVM and BPNN were undertaken. The results, eloquently portrayed in Table 2 and Fig. 4, unravel the outcomes divided into three distinct intervals.

Table 2 The price range recognition result on the eBay dataset when n = 3.

Method	Precision	Recall	F1-score	
SVM	0.889	0.879	0.884	
BPNN	0.891	0.883	0.887	
LSTM	0.903	0.911	0.907	
GRU	0.912	0.907	0.909	
Ours	0.957	0.961	0.959	

Figure 4 The price range recognition result on eBay dataset when n = 3.

Figure 4 unveils a discernible enhancement in precision, specifically at n = 3, as the GRU method undergoes dynamic augmentation through reinforcement learning, transcending from 0.912 to 0.957. Furthermore, the overarching F1-score of the model exhibits noteworthy improvement, underscoring its commendable robustness. Following the meticulous completion of the small interval demarcation (n = 3), the trajectory of our exploration ascends towards higher dimensions, with the division of price intervals into five and eight. The outcomes of this elevated dimensional stratification are artfully depicted in Tables 3 and 4, Figs. 5 and 6.

Table 3 The price range recognition result on the eBay dataset when n = 5.

Method	Precision	Recall	F1-score	
SVM	0.813	0.805	0.809	
BPNN	0.839	0.841	0.840	
LSTM	0.857	0.862	0.859	
GRU	0.892	0.903	0.897	
Ours	0.912	0.891	0.901	

Table 4 The price range recognition result on the eBay dataset when n = 8.

Method	Precision	Recall	F1-score	
SVM	0.806	0.799	0.802	
BPNN	0.811	0.802	0.806	
LSTM	0.837	0.825	0.831	
GRU	0.851	0.847	0.849	
Ours	0.876	0.881	0.878	

Figure 5 The price range recognition result on eBay dataset when n = 5.

Figure 6 The price range recognition result on eBay dataset when n = 8.

The outcomes at five intervals and eight intervals elucidate that the RL-GRU methodology propounded in this treatise continues to manifest superior results across diverse price stratifications. When partitioned into five and eight intervals, its precision in price recognition attains values of 0.912 and 0.876, respectively—outperforming traditional methodologies such as SVM and BPNN by a substantial margin. Notably, akin to the three intervals, the overarching performance undergoes augmentation, attributed to the fortification imparted by reinforcement learning.

A meticulous comparative analysis was orchestrated to delve deeper into the discernible impact of reinforcement learning, specifically DQN, on the GRU methodology. The results, elegantly portrayed in Fig. 7, unveil the nuanced improvements afforded by the integration of reinforcement learning across varying intervals.

Figure 7 The price range recognition comparison result on eBay dataset when n changes.

Figure 7 perceptibly illustrates that concurrent with the escalation of division intervals, the recognition performance of the model posited in this manuscript undergoes a marginal decline yet consistently maintains a commendable threshold surpassing 0.85. Moreover, as the price range expands, the recognition performance exhibits a subtler variation. This nuanced observation suggests that the efficacy of reinforcement learning extends beyond mere augmentation of overall recognition accuracy, encompassing a more conspicuous enhancement in the model’s robustness. For misclassification, the main misclassification objects also appear within adjacent classification intervals, which is necessary to strengthen the model further in the future.

The practical test for the proposed methods

After completing model testing and validation on the eBay dataset, we also analysed and tested the historical data of the auction items we were interested in. For the samples we were interested in, we collected nearly 500 pieces of data, covering all the information needed. The ensuing results, categorised under different price intervals, are meticulously depicted in Figs. 8 and 9.

Figure 8 The price range recognition result on self-established dataset when n = 3.

Figure 9 The price range recognition result on self-established dataset when n = 5.

Figure 8 encapsulates the outcomes of price interval recognition when segmented into three intervals. Discernibly, the RL-GRU methodology outshines its counterparts in recognising this category of auction items, boasting a precision of 0.971 within the three intervals. This achievement significantly surpasses both the GRU and LSTM methods. Furthermore, the results pertaining to recall and F1-score are equally laudable.

Following the meticulous completion of price classification and identification within the ambit of three price intervals, our trajectory advanced towards the classification and identification within five intervals. The results reveal a higher overall recognition rate for the internally curated auction data. This upswing is attributed to the inherent uniformity of goods and the relatively stable regional dynamics, resulting in a smoother price distribution. The recognition precision within the five intervals achieves a commendable 0.953, underscoring a noteworthy efficacy in recognition outcomes.

After rigorous model testing within the five intervals, a meticulous comparative analysis ensued, scrutinising the recognition results across distinct dimensions of price interval division for the LSTM, GRU, and RL-GRU methodologies. The ensuing outcomes are vividly delineated in Fig. 10.

Figure 10 The price range recognition result on self-established dataset when n changes.

Figure 10 shows that the GRU method, standing as a singular entity, outperforms the LSTM method in terms of overall recognition accuracy. Introducing the reinforcement learning model contributes to a discernible enhancement in overall recognition precision. Simultaneously, the overarching performance of the model, fortified by reinforcement learning, experiences an amelioration. It is noteworthy that even under maximum interval division, the results of this amalgamated model surpass those of the singular model.

To better demonstrate the model’s performance, in addition to conducting ablation experiments with different interval predictions, we also analysed the recognition results under different hidden layers, as shown in Fig. 11. In Fig. 11, we can observe that for the recurrent neural network model itself, changing the network depth, i.e., hiding the number of network layers, does not significantly affect model improvement. This article further improved the model’s performance through dynamic environment reinforcement using reinforcement learning DQN.

Figure 11 The price range recognition result with different hidden layers.

Discussion

This study delves into the intricacies of predicting the selling price range in commodity auctions. The aspiration is to forecast the ultimate price range by harnessing the multidimensional attributes intrinsic to auction products, conjoined with features during the auction. An RL-GRU framework, leveraging reinforcement learning and GRU, is introduced to pursue this objective. After conducting crawler-based data analysis on eBay and self-built datasets, the validation results showed that the RL-GRU model predicted online auction product prices well, surpassing traditional methods such as SVM, BPNN, and LSTM. This advantage is mainly due to the flexibility of reinforcement learning in dealing with sequential decision problems; It gradually adjusts its strategy through interaction with the environment to better respond to dynamic changes. In addition, the gating mechanism of GRU enhances the modelling ability of temporal data, enabling the model to capture the trend of price changes flexibly. The end-to-end training method also reduces dependence on complex feature engineering, further improving the accuracy and efficiency of the RL-GRU model in predicting network auction prices. It successfully achieves accurate predictions of different price ranges and satisfactory results.

Applying the RL-GRU method and akin AI models for predicting auctioned commodity prices presents multifaceted advantages. It optimises price decisions, augments pricing accuracy, attracts more prospective buyers, and enhances auction allure and competitiveness, thereby boosting efficiency and transaction completion rates. Furthermore, risk mitigation for auction participants is achieved, providing reliable insights into market trends and commodity price fluctuations.

The technological advancement introduced by this model elevates the auction industry, ushering in smarter tools and methods. Practical application necessitates continual model refinement to enhance data quality and diversity, bolstering the model’s generalisation ability. A broader dataset encompassing diverse commodity types and auction scenarios is imperative. Optimisation for real-time and dynamic auction environments is pivotal to ensuring timely responsiveness to price changes and market volatility. Model interpretability becomes paramount to elucidating decision-making processes, enhancing credibility in practical applications. Tailoring the model’s objective function and strategy aligns with the specific needs of the auction business, fostering a more nuanced adaptation to real-world requirements.

Conclusion

This article delves into the intricate realm of price interval prediction within the auction process, presenting a method grounded in the RL-GRU framework. The methodology commences by constructing model input vectors synthesising quantized features encompassing product attributes, price, auction time, and more. Subsequently, deep features are extracted from these attributes by leveraging the GRU method, resulting a dynamic environmental connection through reinforcement learning. The classification module is then invoked to accomplish the categorization of price ranges. Validation results using the eBay public dataset showcase the robustness of the proposed RL-GRU framework, yielding recognition rates of 0.957, 0.912, and 0.876 across various interval requirements. Furthermore, under a self-curated dataset with three and five price intervals, the recognition precision reaches 0.971 and 0.953, respectively. These outcomes furnish a reference for interval quotations in future auction processes and technical underpinnings for the burgeoning intelligence within this industry.

We plan to use semi-supervised and unsupervised learning methods to divide auction price ranges in future research. These methods will allow us to utilise unlabeled data more effectively. Still, they also bring a series of challenges, including how to ensure data diversity and quality, how to design processing strategies for complex data, and how to verify the accuracy and reliability of models. We will explore solutions suitable for these challenges and rigorously test the practical application effects of the methods to ensure their effectiveness in real-world environments.

Supplemental Information

Supplemental Information 1 Code and dataset.

Additional Information and Declarations

Competing Interests

Author Contributions

Data Availability

The authors declare that they have no competing interests.

Da Ke conceived and designed the experiments, analyzed the data, performed the computation work, prepared figures and/or tables, and approved the final draft.

Xianhua Fan performed the experiments, analyzed the data, performed the computation work, prepared figures and/or tables, authored or reviewed drafts of the article, and approved the final draft.

The following information was supplied regarding data availability:

The code associated with the article is available in the Supplemental File.

The data is available at Zenodo: Ali Hassan Dahir. (2023). Input dataset-Comparative Study of Spatial and Non-spatial Modelling in Price Prediction [Data set]. Zenodo. https://doi.org/10.5281/zenodo.8126348.

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
