# Peer review of "Deep reinforcement learning models in auction item price prediction: an optimisation study of a cross-interval quotation strategy"

_PeerJ Computer Science, doi:10.7717/peerj-cs.2159_

## Round 0.1 · original submission · Major Revisions

Based on the reviewers’ comments, you may resubmit the revised manuscript for further consideration. Please consider the reviewers’ comments carefully and submit a list of responses to the comments along with the revised manuscript.

Reviewer 1 ·

Basic reporting

Overall the work presents a unique RL-GRU paradigm that shows promise in terms of performance and precision over a range of datasets.

The literature study offers a thorough analysis of the use of deep learning techniques in the financial markets, emphasizing how successful they are in trading tactics and portfolio management. The authors skillfully demonstrate the promise of deep learning models, such as neural networks and reinforcement learning, in handling difficult problems in financial decision-making by combining findings from other studies.

The review, however, does not critically analyze the shortcomings and constraints connected to these approaches, which could provide readers with a more unbiased viewpoint. Furthermore, the review's general coherence might be improved by improving the transitions between the various research.

Although the author addressed reinforcement learning strategies in Section 3.2, with a special emphasis on DQN, the authors are urged to include more references to bolster the ideas they describe to make their explanation seem more thorough and credible. Adding more references will bolster the theoretical basis and show how comprehensive the literature review was.

The manuscript's language and writing quality need to be significantly improved. Before resubmitting, the writers are strongly encouraged to improve the manuscript's clarity, coherence, and grammatical accuracy. To properly handle these language-related issues, it is advised to consult a professional editor skilled in academic writing or a fluent English speaker.

Experimental design

The text mostly presents the experimental data that are quantitative in nature. It is suggested that the writers carry out an ablation study in order to enhance the qualitative components of the investigation. The writers can have a better understanding of the effectiveness and resilience of the suggested strategy by methodically examining its performance under various circumstances. Results from an ablation study will be added to the experimental evaluation to make it more thorough.

Validity of the findings

The conclusions derived from Figures 3 are not very strong. To support the findings derived from these numbers, the authors ought to make an effort to offer more thorough justifications or in-depth analysis. Elucidating the interpretations of the results illustrated in Figures 3 will strengthen the study's overall argument and increase its rigour.

The uniqueness brought about by the suggested approach must be clearly defined. The writers are urged to point out specifically how their strategy differs from other approaches in the field. Emphasising the distinct contributions of the suggested approach will highlight its importance in the field of study.

Reviewer 2 ·

Basic reporting

The idea presented in this paper titled “Deep reinforcement learning models in auction item price prediction: an optimization study of a cross-interval quotation strategy” is good. The authors in this paper attempted to predict the prices within the auction domain, introducing a sophisticated RL-GRU framework for price interval analysis. The overall paper is very well written and well structured; however, the authors are suggested to address following comments while revising the paper.
1. The authors are suggested to review and reconsider choice of wording and replace with some simpler synonyms such as “proffer”, “the relentless march of technology” etc.
2. Rewrite the research contributions at the end of introduction section in few words.
3. Highlight the research gaps found from existing literature by adding a paragraph at the end of literature review section. In addition, also highlight what challenges out of being highlighted are being addressed in this study.
4. Extend the literature review section by adding more recent studies.

Experimental design

5. Add a figure at the start of section 3 that illustrates the overall flow of steps involved in the methodology.
6. Increase the size readability and quality of all figures.
7. Add descriptive statistics of datasets used in this study. Also provide the details on the availability of datasets along with sources. Particularly regarding self-curated dataset.
8. Discuss experimental results presented in section 4.2 in more detail. Also add Confusion Matrix.

Validity of the findings

9. Provide the key results in the tabular format as well, that will be a reference for future studies.

Reviewer 3 ·

Basic reporting

This study introduces an advanced RL-GRU framework designed for predicting auction prices. It efficiently extracts features from auction commodities using GRU and employs reinforcement learning for dynamic interactions. The framework accurately divides intervals and identifies prices, achieving precision over 90% across various datasets and intervals. It offers valuable insights for tackling auction price prediction challenges in the future.

The paper is fascinating and can applied in many applications.
- The figures in the manuscript need to be clarified.

Experimental design

1. The aim to augment the sample capacity of the model suggests a recognition of potential limitations in handling larger or more complex datasets. Scaling up the model's capacity may introduce challenges in terms of computational resources, training time, and model performance stability.
2. While the paper mentions the aspiration to explore semi-supervised or unsupervised methodologies for partitioning auction price intervals, it does not delve into the specific challenges or considerations associated with these approaches. Implementing such methodologies may introduce additional complexities and require careful validation to ensure effectiveness.
3. The paper provides an overview of the methodology but lacks detailed insights into the specific parameters, architectures, or hyperparameter tuning processes employed in the RL-GRU framework. Without this information, replicating or extending the proposed approach may be challenging for other researchers.

Validity of the findings

1. The recognition rates are reported across various interval requirements, but the specific characteristics or complexities of these intervals are not detailed. It's possible that the framework performs differently under different interval configurations, and the robustness across a wider range of interval requirements should be investigated

Additional comments

More comparisons with the state-of-the-art models need to be investigated.

---

## Round 0.2 · accepted · Accept

Congratulations, the reviewers are satisfied with the revisions and recommend an accept decision.

Reviewer 1 ·

Basic reporting

Several weeks ago, I had the opportunity to review the original version of this paper. I am pleased to report that the authors have satisfactorily addressed the comments and suggestions from the previous review. The paper is now much stronger and more polished, and I am confident that it will be of interest to a wider audience.

In particular, the authors have:

Clarified the research question and methodology. The authors have now provided a more detailed explanation of their research question and methodology, which makes it easier to understand the significance of their findings.

Strengthened the analysis. The authors have added additional data and analysis to support their claims, which makes their arguments more convincing.

Improved the writing style. The authors have revised the writing to make it more clear and concise, which makes the paper easier to read.

Overall, I am very impressed with the revisions that the authors have made.

Experimental design

Several weeks ago, I had the opportunity to review the original version of this paper. I am pleased to report that the authors have satisfactorily addressed the comments and suggestions from the previous review. The paper is now much stronger and more polished, and I am confident that it will be of interest to a wider audience.

In particular, the authors have:

Clarified the research question and methodology. The authors have now provided a more detailed explanation of their research question and methodology, which makes it easier to understand the significance of their findings.

Strengthened the analysis. The authors have added additional data and analysis to support their claims, which makes their arguments more convincing.

Improved the writing style. The authors have revised the writing to make it more clear and concise, which makes the paper easier to read.

Overall, I am very impressed with the revisions that the authors have made.

Validity of the findings

Several weeks ago, I had the opportunity to review the original version of this paper. I am pleased to report that the authors have satisfactorily addressed the comments and suggestions from the previous review. The paper is now much stronger and more polished, and I am confident that it will be of interest to a wider audience.

In particular, the authors have:

Clarified the research question and methodology. The authors have now provided a more detailed explanation of their research question and methodology, which makes it easier to understand the significance of their findings.

Strengthened the analysis. The authors have added additional data and analysis to support their claims, which makes their arguments more convincing.

Improved the writing style. The authors have revised the writing to make it more clear and concise, which makes the paper easier to read.

Overall, I am very impressed with the revisions that the authors have made.

Reviewer 2 ·

Basic reporting

Thank you for making efforts in addressing the raised comments.

Experimental design

no comment

Validity of the findings

no comment

Reviewer 3 ·

Basic reporting

The research was well-presented.

Experimental design

Well.

Validity of the findings

The conclusion is appropriate.

Additional comments

This research has been revised and improved appropriately based on the provided feedback. There are no further suggestions for improvement. It is deemed suitable for acceptance.